# A novel genotype of *Hantaan orthohantavirus* harbored by *Apodemus agrarius chejuensis* as a potential etiologic agent of hemorrhagic fever with renal syndrome in Republic of Korea

**Kyungmin Park**[1,2], **Won-Keun Kim**[3,4], **Seung-Ho Lee**[2], **Jongwoo Kim**[2], **Jingyeong Lee**[2], **Seungchan Cho**[2], **Geum-Young Lee**[2], **Jin Sun No**[2,5], **Keun Hwa Lee**[6], **Jin-Won Song**[1,2]*

**1** BK21 Graduate Program, Department of Biomedical Sciences, Korea University College of Medicine, Seoul, Republic of Korea, **2** Department of Microbiology, Korea University College of Medicine, Seoul, Republic of Korea, **3** Department of Microbiology, College of Medicine, Hallym University, Chuncheon, Republic of Korea, **4** Institute of Medical Research, College of Medicine, Hallym University, Chuncheon, Republic of Korea, **5** Division of High-risk Pathogens, Bureau of Infectious Diseases Diagnosis Control, Korea Disease Control and Prevention Agency, Cheongju, Republic of Korea, **6** Department of Microbiology, College of Medicine, Hanyang University, Seoul, Republic of Korea

☉ These authors contributed equally to this work.

* jwsong@korea.ac.kr

**Data Availability Statement:** The mitochondrial DNA sequences were deposited in GenBank (accession numbers: MW219774- MW2197749).

## Abstract

### Background

Orthohantaviruses, causing hemorrhagic fever with renal syndrome (HFRS) and hantavirus cardiopulmonary syndrome, pose a significant public health threat worldwide. Despite the significant mortality and morbidity, effective antiviral therapeutics for orthohantavirus infections are currently unavailable. This study aimed to investigate the prevalence of HFRS-associated orthohantaviruses and identify the etiological agent of orthohantavirus outbreaks in southern Republic of Korea (ROK).

### Methodology/Principal findings

We collected small mammals on Jeju Island during 2018–2020. We detected the Hantaan virus (HTNV)-specific antibodies and RNA using an indirect immunofluorescence assay test and reverse transcription-polymerase chain reaction on *Apodemus agrarius chejuensis* (*A. chejuensis*). The prevalence of anti-HTNV antibodies among rodents was 14.1%. A total of six seropositive mouse harbored HTNV RNA. The amplicon-based next-generation sequencing provided nearly full-length tripartite genomic sequences of six HTNV harbored by *A. chejuensis*. Phylogenetic and tanglegram analyses were conducted for inferring evolutionary relationships between orthohantaviruses with their reservoir hosts. Phylogenetic analysis showed a novel distinct HTNV genotype. The detected HTNV genomic sequences were phylogenetically related to a viral sequence derived from HFRS patient in southern ROK. Tanglegram analysis demonstrated the segregation of HTNV genotypes corresponding to *Apodemus spp.* divergence.

The genomic sequences of HTNV were deposited in GenBank (accession numbers: MW219756-MW219773).

**Funding:** This study was supported by the Research Program to Solve Social Issues of the National Research Foundation of Korea (NRF) funded by the Ministry of Science and Information and Communication Technology (ICT) (NRF-2017M3A9E4061992 to J.-W.S.) and the Agency for Defense Development (UE202026GD to J.-W.S.). The funders had no role in study design, data collection and analysis, decision to publish, or preparation of the manuscript.

**Competing interests:** The authors declare no conflict of interest.

## Conclusions/Significance

Our results suggest that *A. chejuensis*-borne HTNV may be a potential etiological agent of HFRS in southern ROK. Ancestral HTNV may infect *A. chejuensis* prior to geological isolation between the Korean peninsula and Jeju Island, supporting the co-evolution of orthohantaviruses and rodents. This study arises awareness among physicians for HFRS outbreaks in southern ROK.

### Author summary

According to the Korea Disease Control and Prevention Agency, about 18 HFRS cases have been reported over the past decade on Jeju Island, Republic of Korea (ROK). Currently, no report for the etiological agent associated with hemorrhagic fever with renal syndrome (HFRS) on Jeju Island is available. We report the nearly whole-genome sequences and have characterized a novel HTNV genotype carried by the Jeju striped field mice (*Apodemus chejuensis*) collected from Jeju Island. The phylogenetic analysis of HTNV revealed well-supported phylogeographic clusters, showing a high genetic divergence by the distribution of rodent reservoir hosts. To our knowledge, this study is the first provisional report that suggests molecular evidence of orthohantavirus outbreaks associated with hantaviral disease on Jeju Island. Although our observations are provisional, the novel genotype of HTNV harbored by *A. chejuensis* may be a potential etiological agent of HFRS in southern ROK. This study provides important insights into the male-predominance prevalence, genetic diversity, and molecular evolutionary dynamics of orthohantaviruses circulating in the ROK.

## Introduction

*Orthohantavirus* (Family *Hantaviridae*, Order *Bunyavirales*) is an enveloped negative-sense single-stranded RNA virus containing large (L), medium (M), and small (S) genome segments [1]. The tripartite segments encode an RNA-dependent RNA polymerase (RdRp), two membrane glycoproteins ($G_N$ and $G_C$), and a nucleocapsid (N) protein. Orthohantaviruses are etiological agents of hemorrhagic fever with renal syndrome (HFRS) with fatality rates of 1–15% in Eurasia [2]. Natural reservoirs of orthohantaviruses belong to various small mammals, including rodents (Rodentia), bats (Chiroptera), and insectivores (Soricomorpha) [3–5]. HFRS is mainly caused by Hantaan virus (HTNV), Seoul virus (SEOV), Dobrava virus (DOBV), and Puumala virus [6]. Orthohantaviruses are transmitted to humans by inhaling aerosolized infectious particles derived from saliva, urine, and feces of infected rodents. Antiviral therapeutics for orthohantavirus infections remain ineffective and unavailable despite the significant mortality and morbidity of the diseases.

Emerging orthohantaviruses pose a significant public health threat worldwide. According to the Korea Disease Control and Prevention Agency, approximately 400 HFRS cases are recorded annually in the Republic of Korea (ROK), with a mean mortality rate of 1–4% [7]. In 1976, HTNV harbored by the striped field mice (*Apodemus agrarius*), was first identified in the ROK [8]. In addition, five types of hantaviruses have been characterized: SEOV carried by *Rattus norvegicus and R. rattus*; Soochong virus (SOOV) carried by *A. peninsulae*; Muju virus carried by *Myodes regulus*; Imjin virus carried by *Crocidura lasiura*, and Jeju virus carried by

*C. shantungensis* [9–13]. Previous studies have reported genomic and phylogeographic relationships between the patients with HFRS and *A. agrarius* collected in endemic regions [14]. However, most epidemiological surveillance studies have been performed in the northern ROK, which limits the understanding of the correlation between HFRS clinical cases and their reservoirs in southern areas [15–18]. Although a few studies have demonstrated the prevalence of orthohantavirus in the southern ROK, epidemiological investigations of orthohantavirus outbreaks in humans and rodents in the southern ROK were limited due to the lack of available viral sequences [19–23]. On Jeju Island, an administrative island in southern ROK, approximately 18 HFRS cases have been reported from 2011 to 2019 [7]. Based on the incidence of HFRS, one or more distinct, rodent-borne orthohantaviruses have been suspected to exist on the island. However, the etiological agent of HFRS on Jeju Island still remains unknown.

In this study, we report the nearly full-length genome sequences and characterize a genetically distinct orthohantavirus, harbored by the subspecies *A. agrarius chejuensis* (*A. chejuensis*) captured on Jeju Island. The phylogenetic association of *A. chejuensis*-borne HTNV with viral sequences from HFRS patient suggests that *A. chejuensis*-borne HTNV may be a potential etiological agent of HFRS on Jeju Island, ROK. This study provides significant insights into the genetic diversity and molecular evolution of HTNV in the ROK. We highlight the necessity of continued surveillance and large-scale investigations for understanding the evolutionary diversification, transmission, and pathogenicity of orthohantaviruses in the southern areas, ROK.

## Results

### Trapping for small mammals on Jeju Island, ROK

Rodents and shrews were captured on Jeju Island, ROK from 2018 to 2020 (**Fig 1**). The trapped small mammals consisted of 91.4% (64/70) *A. chejuensis*, 5.7% (4/70) *C. shantungensis*, and 2.9% (2/70) *Tscherskia triton* (**Table 1**). *A. chejuensis* represented 91.4% of all individuals trapped on Jeju Island, and the most frequently captured species at all trapping sites. *A. chejuensis* captured on Jeju Island was genetically distinct from *A. agrarius* captured in mainland, ROK (**S1 Fig**).

### Serological and molecular prevalence for HTNV collected on Jeju Island

A total of 9/64 (14.1%) rodents were seropositive, of which 1/12 (8.3%), 1/5 (20%), 3/7 (42.9%), 1/19 (5.3%), and 3/15 (20%) were detected in Ara-dong, Aewol-eup, Bongseong-ri, Haengwon-ri, and Ora-dong, respectively, with no seropositive rodents being detected in Yongsu-ri (**Table 2 and S2 Fig**). The prevalence of anti-HTNV antibodies was 7/35 (20%) in males and 2/29 (6.9%) in females. HTNV-specific reverse transcription-polymerase chain reaction (RT-PCR) was conducted for viral RNA. In total, 6/9 (66.7%) seropositive *A. chejuensis* harbored HTNV RNA, consisting of 5/7 (71.4%) males and 1/2 (50%) females. The antibodies against orthohantaviruses were not detected in *C. shantungensis* and *T. triton*.

### HTNV RNA loads in tissues from *A. chejuensis*

The HTNV RNA load in *A. chejuensis* was determined by quantitative PCR (qPCR) using various samples from the brains, lungs, livers, kidneys, spleens, reproductive organs, and hearts (**Fig 2**). Both IFA$^+$PCR$^-$ and IFA$^-$PCR$^-$ groups showed that HTNV RNA was undetectable in all tissues. In IFA$^+$PCR$^+$ rodents, viral RNA was identified in multiple tissues from rodents (Ac19-6, Ac20-5, Ac20-6, Ac20-30, Ac20-31, and Ac20-32). The lung tissues harbored the highest amount of HTNV RNA (Ct values 21.8–31.5) in all evaluated rodents except Ac20-32.

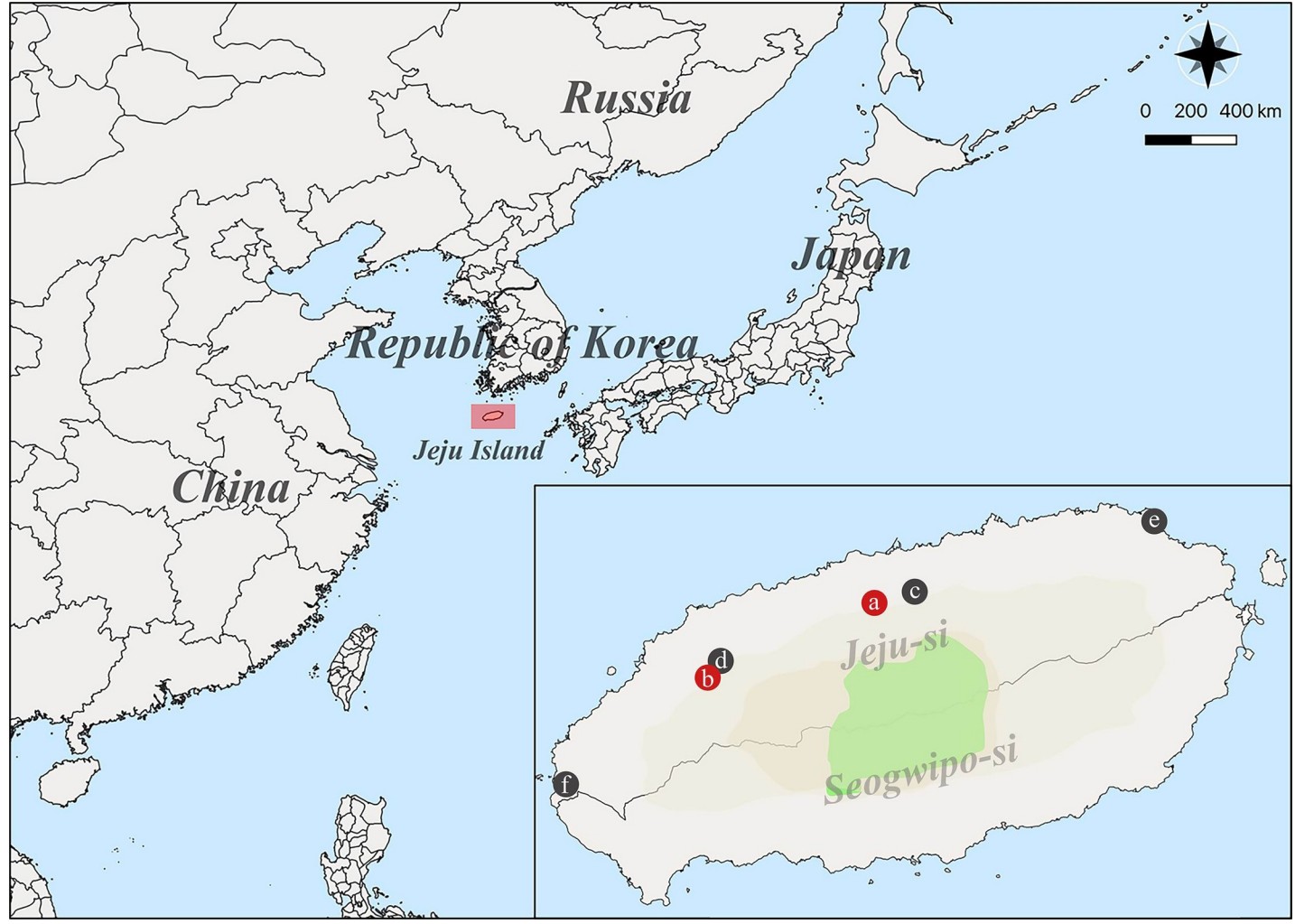

**Fig 1. Geography of trapping sites of Hantaan virus (HTNV) collected on Jeju Island, the Republic of Korea.** A geographic map shows different trapping locations where small mammals were captured on Jeju Island from 2018 to 2020. The red circles indicate the HTNV RNA positive areas: a, Ora-dong; b, Bongseong-ri in Jeju-si, respectively. The black circle represents the regions where no HTNV RNA was identified: c, Ara-dong; d, Aewol-eup; e, Haengwon-ri; f, Youngsu-ri. The Quantum Geographical Information System (QGIS) 3.10 for Mac was used to create the map.

The brain tissues showed the presence of HTNV RNA in all rodents except Ac20-6. The rodents (Ac20-5, Ac20-31, and Ac20-32) contained the HTNV RNA in reproductive organs.

## Whole-genome sequencing of HTNV

Using multiplex PCR-based next-generation sequencing (NGS), the nearly whole-genome sequence of HTNV was obtained from the lung tissues of *A. chejuensis* captured on Jeju Island.

**Table 1. Trapping results of small mammals collected at sites on Jeju Island in the Republic of Korea, 2018–2020.**

| Species | Ara-dong | Aewol-eup | Bongseong-ri | Haengwon-ri | Ora-dong | Yongsu-ri | Total (%) |
|---|---|---|---|---|---|---|---|
| *Apodemus chejuensis* | 12 | 5 | 7 | 19 | 15 | 6 | 64 (91.4) |
| *Crocidura shantungensis* | - | - | - | 2 | 1 | 1 | 4 (5.7) |
| *Tscherskia triton* | - | - | - | 2 | - | - | 2 (2.9) |
| Total | 12 | 5 | 7 | 23 | 16 | 7 | 70 (100) |

**Table 2. Serological and molecular prevalence of Hantaan virus (HTNV) in *Apodemus chejuensis* captured on Jeju Island from 2018 to 2020.**

| Trapping site | Number of captured *A. chejuensis* | Seropositivity for anti-HTNV IgG (%) | | | HTNV RNA positivity (%) | | |
|---|---|---|---|---|---|---|---|
| | | Male | Female | Total | Male | Female | Total |
| Ara-dong | 12 | 1/5 (20) | 0/7 | 1/12 (8.3) | 0/1 | -[a] | 0/1 |
| Aewol-eup | 5 | 1/3 (33.3) | 0/2 | 1/5 (20) | 0/1 | - | 0/1 |
| Bongseong-ri | 7 | 2/4 (50) | 1/3 (33.3) | 3/7 (42.9) | 2/2 (100) | 1/1 (100) | 3/3 (100) |
| Haengwon-ri | 19 | 0/12 | 1/7 (14.3) | 1/19 (5.3) | - | 0/1 | 0/1 |
| Ora-dong | 15 | 3/9 (33.3) | 0/6 | 3/15 (20) | 3/3 (100) | - | 3/3 (100) |
| Yongsu-ri | 2 | 0/2 | 0/4 | 0/6 | - | - | - |
| Total | 64 | 7/35 (20) | 2/29 (6.9) | 9/64 (14.1) | 5/7 (71.4) | 1/2 (50) | 6/9 (66.7) |

[a]; Seronegative sample was not analyzed by RT-PCR

**Table 3** shows that 41.8–93.8%, 97.3–99.9%, and 99.3–99.7% coverages of the L, M, and S segments of six *A. chejuensis* borne-HTNV. The average number of viral reads and depth of coverage is shown in the **S1 Table**. The uncovered sequences of HTNV tripartite RNA were determined from HTNV Ac19-6, Ac20-5, Ac20-6, Ac20-30, Ac20-31, and Ac20-32, using conventional RT-PCR. To complete the whole-genome sequence of HTNV, 5′ and 3′ end rapid amplification of cDNA ends (RACE) PCR was performed in HTNV Ac20-30. The 5′ termini genome sequences of HTNV Ac20-30 were recovered in tripartite segments, while the 3′ end viral sequence was obtained only in the M segment. The 5′ end sequences of HTNV were 5′-GGA GUC UAC UAC UA-3′ in the tripartite segments. The 3′ termini sequence of the HTNV M segment was 5′-UAG UAG UAU GCU CC-3′, showing incomplete complementarity due to a mismatch at position 9 and a noncanonical U–G pair at position 10.

## Genomic characterization of HTNV

The nucleotide identity of the HTNV L, M, and S segments from Jeju Island showed 82.2–83.1%, 87.2–88.8%, and 84.8–85.4%, respectively, compared to those of HTNV 76–118 (**S2–S4 Tables**). The amino acid sequences differed between HTNV from Jeju Island and HTNV 76–118 by 2.8–3.9%, 1.8–2.5%, and 3.3–3.7%. The entire M segment of HTNV harbored by *A. chejuensis* was 3,618 nucleotides with a glycoprotein precursor of 1,135 amino acids, with the highly conserved pentapeptide WAASA motif being found at amino acid positions 644–648.

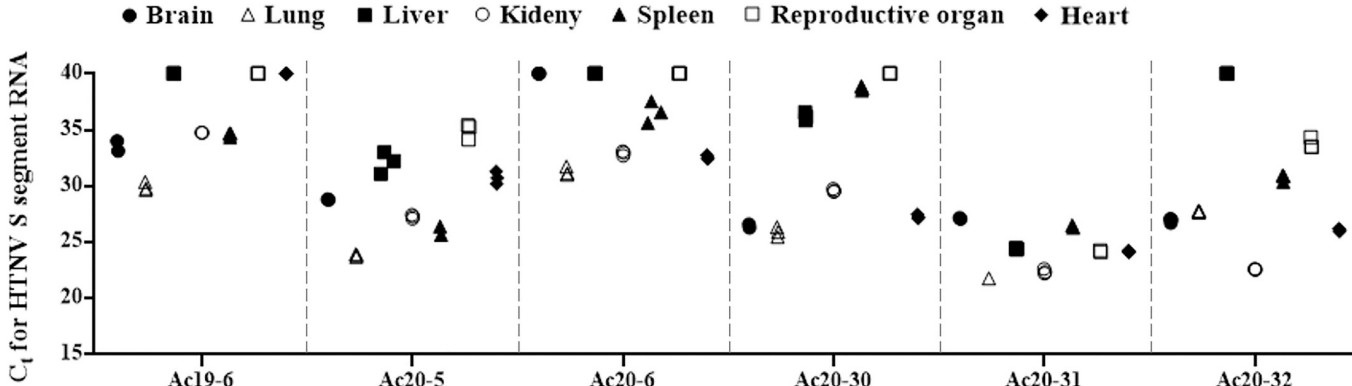

**Fig 2. Measurement of HTNV RNA loads in different tissues of *Apodemus chejuensis*, Jeju Island during 2018–2020.** Ct values were determined for HTNV RNA of the S segment in brain, lung, liver, kidney, spleen, reproductive organ, and heart tissues obtained from six rodents positive for HTNV IgG and HTNV RNA. Ct, cycle threshold; S, small; HTNV, Hantaan virus; IgG, immunoglobulin G.

**Table 3. Next-generation sequencing coverages of Hantaan virus (HTNV) from rodents collected in Jeju Island, 2018–2020.**

| Site | Sample | IFA for anti-HTNV IgG | Origin | Ct value | HTNV genomes, % coverage | | |
|---|---|---|---|---|---|---|---|
| | | | | | L segment | M segment | S segment |
| Ora-dong, Jeju-si | Ac19-6 | 1:1,024[a] | Lung | 30.0 | 81 | 96.6 | 98.1 |
| | Ac20-5 | 1:32[b] | Lung | 23.8 | 58.6 | 99.2 | 98.4 |
| | Ac20-6 | 1:16[b] | Lung | 31.5 | 41.4 | 97 | 98.4 |
| Bongseong-ri, Jeju-si | Ac20-30 | 1:256[a] | Lung | 25.9 | 93.4 | 99.1 | 98.4 |
| | Ac20-31 | 1:256[a] | Lung | 21.8 | 93 | 99 | 98.4 |
| | Ac20-32 | 1:8[b] | Lung | 27.7 | 91 | 99.1 | 98.4 |

Ac, *Apodemus chejuensis*; IFA, indirect immunofluorescence antibody test; IgG, immunoglobulin G; $C_t$, cycle threshold; L, large; M, medium; S, small.

[a]; IFA test was performed from sera

[b]; IFA test was performed from heart fluids.

Genome coverage was calculated by obtaining consensus sequences matching the prototype HTNV 76–118.

The complete genome sequence lengths of HTNV are the L (6,533 nt), M (3,616 nt), and S segments (1,696 nt), respectively.

HTNV strains from *A. chejuensis* showed seven potential N-linked glycosylation sites in two glycoproteins: five in Gn (N134, N235, N347, N399, and N609) and two in Gc (N766 and N928) (**S3 Fig**). The glycosylation sites were identical to those of HTNV 76–118 harbored by *A. agrarius*.

## Phylogenetic and co-evolutionary analyses of HTNV

The phylogenetic analysis of the HTNV L, M, and S segments showed well-supported geographical clusters (**Fig 3**). The L segment of HTNV from Jeju Island formed an independent geographic and distinct cluster with all other viral genomes collected from mainland ROK and China. The M segment of HTNV from Jeju Island shared a common ancestor with the HTNV from mainland ROK. The S segment of HTNV from Jeju Island formed an independent genetic lineage with other HTNV, SOOV, and Amur virus (AMRV). In addition, the phylogenetic analysis of the partial HTNV L, M, and S segments revealed well-supported geographical clusters (**Fig 4**). In all segments, *A. chejuensis*-borne HTNV from Jeju Island formed homologous genetic clades with CUH15-126 obtained from the patients with HFRS in Jangheung-gun (Jeollanam Province) and distinguished from other *A. agrarius*-borne HTNV (Gyeonggi and Gangwon Provinces). Tanglegram analysis showed the segregation of representative rodent-borne hantaviruses according to the species of their reservoir hosts (**Fig 5**). The phylogenetic positions of HTNV from *A. chejuensis* and other hantaviruses mirrored the co-evolutionary relationships of their *Rodentia* hosts, except for Tula and Lúxī virus.

## Genetic reassortment analysis of HTNV

Recombination Detection Program 4 (RDP4) and Graph incompatibility based Reassortment Finder (GiRaF) analyses were performed to evaluate the possibility of genetic reassortment events. The genetic exchange event was undetectable among HTNV genomic sequences in this study.

## Discussion

Hantaviruses have specifically evolved along with one or a few closely related rodent species [4,5,24]. The genus *Apodemus* (family *Muridae*, order *Rodentia*) consists of at least 20 small rodent species and is mainly widespread in the Palearctic area [25]. Molecular data based on

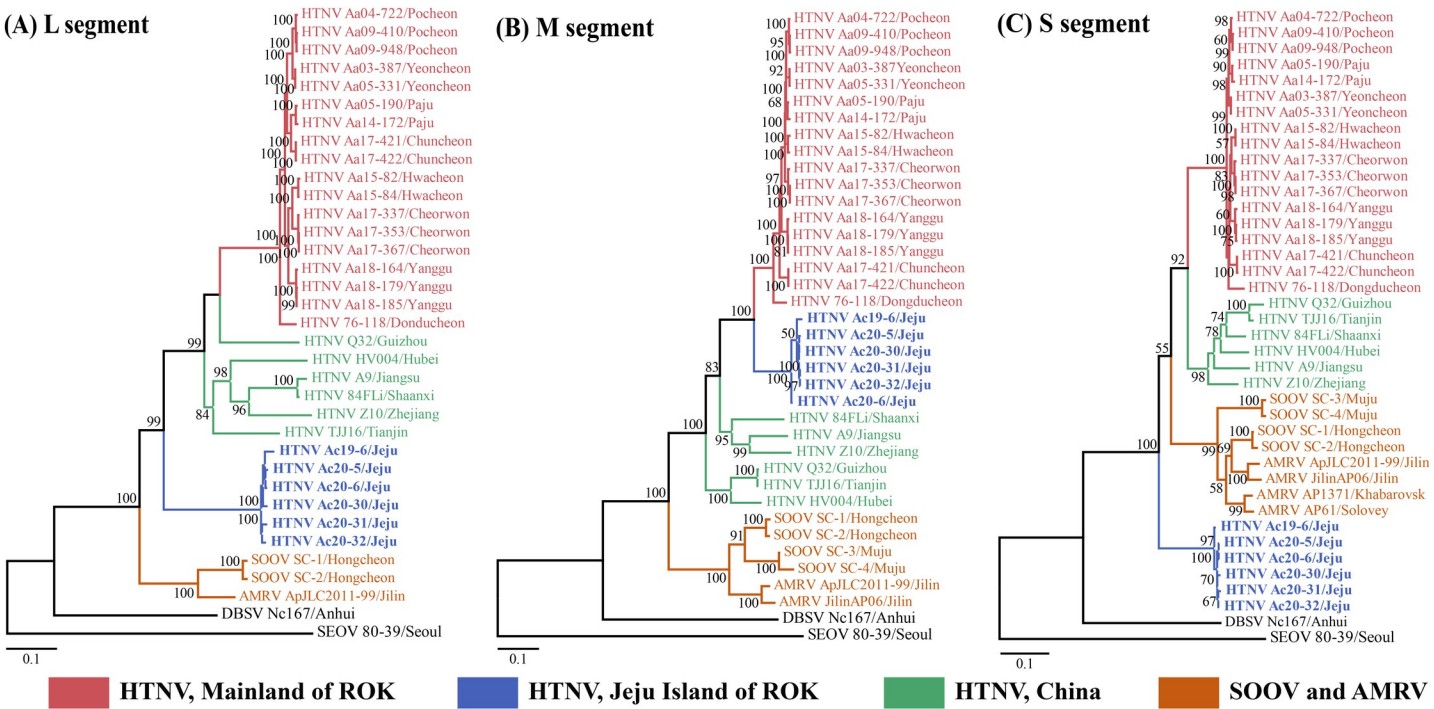

**(A) L segment    (B) M segment    (C) S segment**

**HTNV, Mainland of ROK    HTNV, Jeju Island of ROK    HTNV, China    SOOV and AMRV**

**Fig 3. Phylogeographic analysis of the L, M, and S segments of Hantaan virus (HTNV) collected from rodents in Jeju Island, Republic of Korea (ROK).** Whole-genome sequences of HTNV from *A. chejuensis* of Jeju Island were obtained by multiplex polymerase chain reaction-based next-generation sequencing during 2018–2020. The phylogenies were generated by maximum likelihood methods in MEGA 7 with bootstrap 1,000 iterations based on the HTNV (A) L (1–6,533 nt), (B) M (1–3,618 nt), and (C) S (1–1,696 nt), segments, respectively. The branch lengths were proportional to the number of nucleotide substitutions, while vertical distances were used for clarity. The numbers at each node are bootstrap probabilities determined for 1,000 iterations. HTNV groups are color-coded according to geographic sites (red, mainland ROK; blue, Jeju Island of ROK; green, China). The brown color indicates SOOV and AMRV strains from East Asia (China, Russia, and ROK). The following hantaviral sequences were included in the analysis: HTNV Aa03-387 (L segment, KT934958; M segment, KT934992; S segment, KT935026), Aa04-722 (L segment, KU2071740; M segment, KU207182; S segment, KU207190), Aa05-190 (L segment, KT934959; M segment, KT934993; S segment, KT935027), Aa05-331 (L segment, KT934962; M segment, KT934996; S segment, KT935030), Aa09-410 (L segment, KU207177; M segment, KU207185; S segment, KU207193), Aa09-948 (L segment, KT934966; M segment, KT935000; S segment, KT935034), Aa14-172 (L segment, KT934974; M segment, KT935008; S segment, KT935042), Aa15-82 (L segment, MT012572; M segment, MT012560; S segment, MT012548), Aa15-84 (L segment, MT012573; M segment, MT012561; S segment, MT012549), Aa17-337 (L segment, MT012574; M segment, MT012562; S segment, MT012550), Aa17-353 (L segment, MT012575; M segment, MT012563; S segment, MT012551), Aa17-367 (L segment, MT012576; M segment, MT012564; S segment, MT012552), Aa17-421 (L segment, MT012577; M segment, MT012565; S segment, MT012553), Aa17-422 (L segment, MT012578; M segment, MT012566; S segment, MT012554), Aa18-164 (L segment, MT012579; M segment, MT012567; S segment, MT012555), Aa18-179 (L segment, MT012580; M segment, MT012568; S segment, MT012556), Aa18-185 (L segment, MT012581; M segment, MT012569; S segment, MT012557), Ac19-6 (L segment, MW219756; M segment, MW219762; S segment, MW219768), Ac20-5 (L segment, MW219757; M segment, MW219763; S segment, MW219769), Ac20-6 (L segment, MW219758; M segment, MW219764; S segment, MW219770), Ac20-30 (L segment, MW219759; M segment, MW219765; S segment, MW219771), Ac20-31 (L segment, MW219760; M segment, MW219766; S segment, MW219772), Ac20-32 (L segment, MW219761; M segment, MW219767; S segment, MW219773), 76–118 (L segment, NC_005222; M segment, M14627; S segment, M14626), HV004 (L segment, JQ083393; M segment, JQ083394; S segment, JQ093395), A9 (L segment, AF293665; M segment, AF035831; S segment, AF329390), Q32 (L segment, DQ371906; M segment, DQ371905; S segment, AB027097), 84FLi (L segment, AF336826; M segment, AF345636; S segment, AF366568), Z10 (L segment,NC_006435; M segment, NC_006437; S segment, AF366568), TJJ16 (L segment, KU215675; M segment, EU074672; S segment, AY839871), SOOV SC-1 (L segment, DQ056292; M segment, AY675353; S segment, AY675349), SC-2 (L segment, AY675354; M segment, DQ056293; S segment, AY675350), SC-3 (M segment, DQ056294; S segment, AY675351), SC-4 (M segment, DQ056295; S segment, AY675352), AMRV strains ApJLC2011-99 (L segment, JX473002; M segment, JX473003; S segment, JX473004), JilinAP06 (M segment, EF371454; S segment, EF121324), AP1371 (S segment, AF427324), AP61 (S segment, AB071183), DBSV Nc167 (L segment, DQ989237; M segment, AB027115; S segment, AB027523), and SEOV 80–39 (L segment, NC_005238; M segment, NC_005237; S segment, NC_005236). Abbreviations: SOOV, Soochong virus; AMRV, Amur virus; DBSV, Dàbiéshān virus; SEOV, Seoul virus.

the rodent mitochondrial DNA cytochrome *b* gene and the interphotoreceptor retinoid binding protein gene in nuclear DNA revealed that *Apodemus spp.* diverged approximately 8–10 million years ago [26]. In Asia, *A. agrarius* was first identified as the primary rodent reservoir of HTNV [8]. *A. peninsulae* was also a rodent reservoir for SOOV (closely related to AMRV) distributed in ROK, China, and Russia [10,27,28]. The striped field mouse (*A. agrarius coreae*), the most abundant rodent species, are widely distributed in the Korean peninsula, whereas *A. agrarius chejuensis* is mainly found on Jeju Island, the southernmost island of ROK [29].

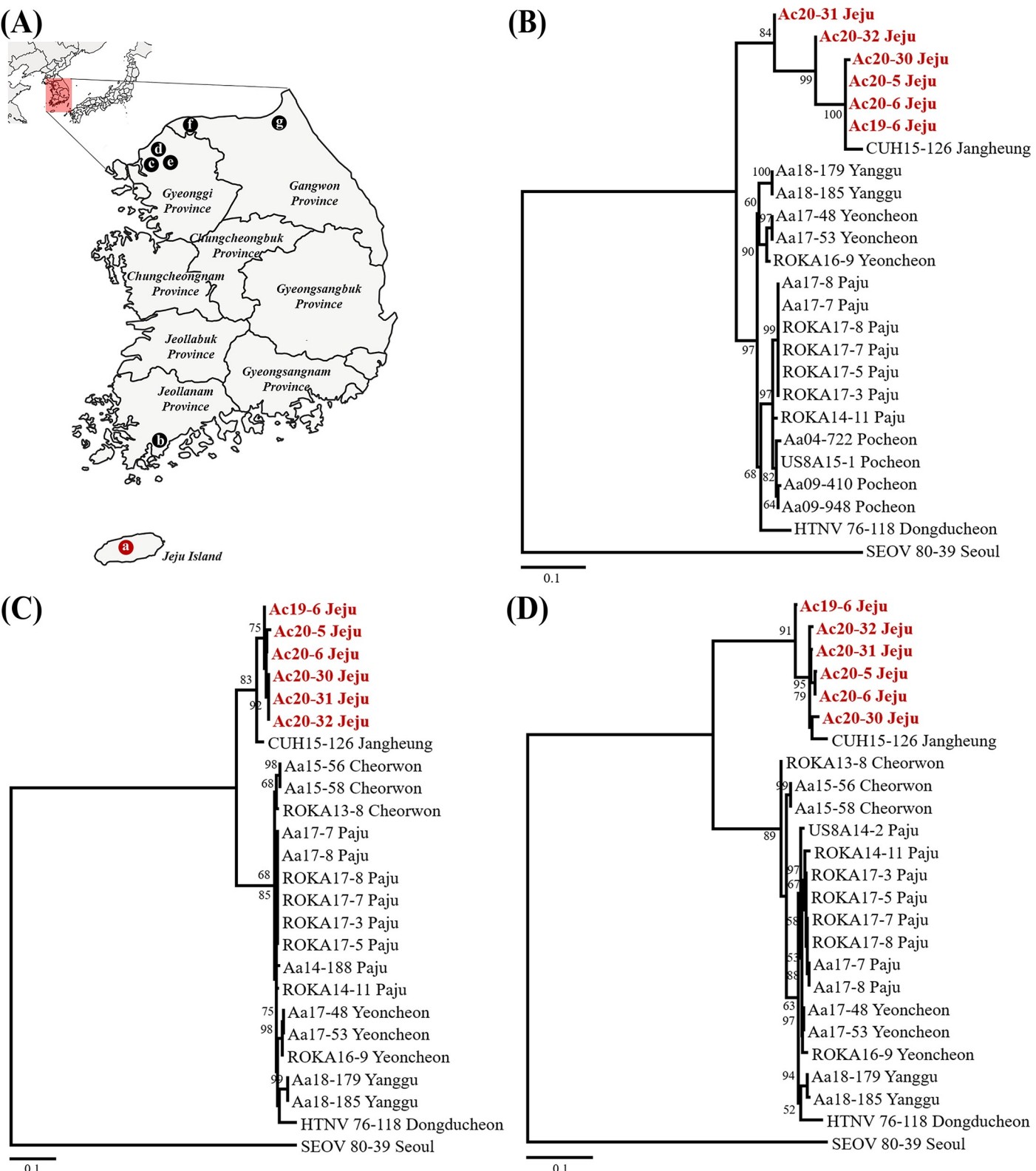

**Fig 4. Phylogeographic analysis of the L, M, and S segments of Hantaan virus (HTNV) collected from rodents and patients with hemorrhagic fever with renal syndrome (HFRS) in Republic of Korea (ROK).** (A) Geographic locations (circle symbols) indicate where the HTNV genomes were acquired in mainland and Jeju Island, ROK. The circle indicates geographic sites (a, Jeju; b, Jangheung; c, Paju; d, Yeoncheon; e, Pocheon; f, Cheorwon; g, Yanggu). Phylogenesis were generated by maximum-likelihood based on the alignment (Clustal W method) using the (B) 353-nucleotide L (coordinates 2,963–3,315 nt), (C) 369-nucleotide M (coordinates

1,973–2,341 nt), and (D) 639-nucleotide S (coordinates 412–1,050 nt) segments of HTNV. Topologies were evaluated using bootstrap analysis of 1,000 iterations. The following viral sequences were included in the analysis: HTNV ROKA13-8 (M segment, KU207202; S segment, KU207206), ROKA14-11 (L segment, KU207199; M segment, KU207203; S segment, KU207207), ROKA16-9 (L segment, MH598466; M segment, MH598480; S segment, MH598494), ROKA17-3 (L segment, MH598467; M segment, MH598481; S segment, MH598495), ROKA17-5 (L segment, MH598468; M segment, MH598482; S segment, MH598496), ROKA17-7 (L segment, MH598469; M segment, MH598483; S segment, MH598497), ROKA17-8 (L segment, MH598470; M segment, MH598484; S segment, MH598498), US8A14-2 (S segment, KU207208), US8A15-1 (L segment, KU207201), CUH15-126 (L segment, MG663537; M segment, MG663538; S segment, MG663539), Aa04-722 (L segment, KU2071740), Aa09-410 (L segment, KU207177), Aa09-948 (L segment, KT934966), Aa14-188 (M segment, KT935009), Aa15-56 (M segment, KU207187; S segment, KU207195), Aa15-58 (M segment, KU207188; S segment, KU207196), Aa17-7 (L segment, MH598475; M segment, MH598489; S segment, MH598503), Aa17-8 (L segment, MH598476; M segment, MH598490; S segment, MH598504), Aa17-48 (L segment, MH598477; M segment, MH598491; S segment, MH598505), Aa17-53 (L segment, MH598479; M segment, MH598493; S segment, MH598507), Aa18-179 (L segment, MT012580; M segment, MT012568; S segment, MT012556), Aa18-185 (L segment, MT012581; M segment, MT012569; S segment, MT012557), Ac19-6 (L segment, MW219756; M segment, MW219762; S segment, MW219768), Ac20-5 (L segment, MW219757; M segment, MW219763; S segment, MW219769), Ac20-6 (L segment, MW219758; M segment, MW219764; S segment, MW219770), Ac20-30 (L segment, MW219759; M segment, MW219765; S segment, MW219771), Ac20-31 (L segment, MW219760; M segment, MW219766; S segment, MW219772), Ac20-32 (L segment, MW219761; M segment, MW219767; S segment, MW219773), 76–118 (L segment, NC_005222; M segment, M14627; S segment, M14626), and SEOV 80–39 (L segment, NC_005238; M segment, NC_005237; S segment, NC_005236). Abbreviations; ROKA, Republic of Korea Army; US8A, United States 8th Army forces; CUH, Chosun University Hospital; Aa, *Apodemus agrarius*; Ac, *A. chejuensis*; SEOV, Seoul virus.

However, a population of *A. agrarius* that are genetically and morphologically clustered with *A. chejuensis* has been reported in Wan Island (80 km North from Jeju Island) located on the southern coast of ROK [30]. Jeju Island was formed by a series of volcanic activities about 1.2 million years ago [31–34]. The island was connected to the Korean peninsula during the Pleistocene and separated about 10,000 years ago. The ancestral population of *A. chejuensis* on Jeju Island may have originated from the Korean peninsula or eastern China [35]. Han et al. and Koh et al. estimated the population of *A. agrarius* and *A. chejuensis* diverged approximately 12 million years ago and 7,000–500,000 years ago (Pleistocene period), respectively, although both studies referred to the last ice age as a crucial period [30,36]. *A. agrarius* and *A. chejuensis* have recently been suggested to be separate genera of *Apodemus spp.* based on mitochondrial genome analysis, morphological characteristics, and crossbreeding experiments [30,36,37].

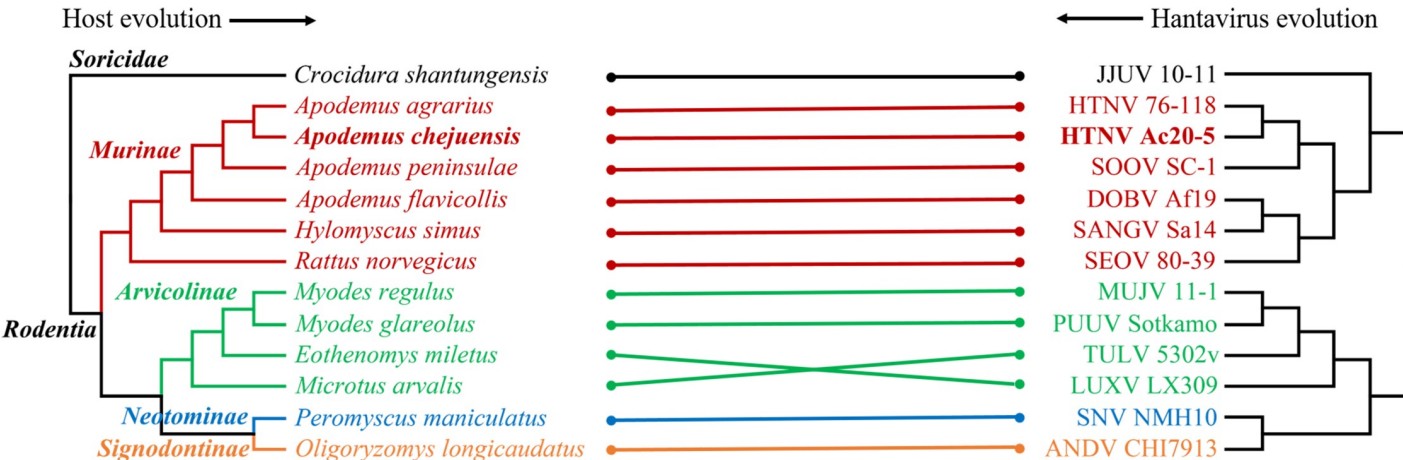

**Fig 5. Tanglegram comparing the phylogenies between hantaviruses and their reservoir hosts in Rodentia.** Tanglegram was generated using the R package, using consensus maximum likelihood topologies based on the nucleotide sequences of the hantaviral M segment (right panel) with mitochondrial DNA cytochrome *b* gene sequences of the reservoir host species (left panel). Letters for taxa are shown in red for *Murinae*, green for *Arvicolinae*, blue for *Signodontinae*, orange for *Neotominae*, black for *Soricidae* in left panel. *Apodemus chejuensis*-borne HTNV (Ac20-5) is shown in bold lettering in each panel. The host species and viruses accession number followed up; reservoir hosts *Crocidura shantungensis* (HQ993053), *A. agrarius* (MN205319), *A. chejuensis* (MW219775), *A. peninsulae* (AB073811), *A. flavicollis* (JF819967), *Hylomyscus simus* (DQ212188), *Rattus norvegicus* (AB355903), *Myodes regulus* (NC_016427), *Myodes glareolus* (AF119272), *Eothenomys miletus* (AY426686), *Microtus arvalis* (EU439459), *Peromyscus maniculatus* (AF119261), *Oligoryzomys longicaudatus* (AF346566), hantavirus strains JJUV 10–11 (NC_034404), HTNV 76–118 (M14627), HTNV Ac20-5 (MW219763), SOOV SC-1 (AY675353), DOBV Af19 (NC_005234), SANGV Sa14 (NC_034516), SEOV 80–39 (NC_005237), MUJV 11–1 (JX028272), PUUV Sotkamo (HE801634), TULV 5302v (NC_005228), LUXV LX3009 (NC_038528), SNV NMH10 (NC_005215), ANDV CHI7913 (AY228238). JJUV, Jeju virus; HTNV, Hantaan virus; SOOV, Soochong virus; DOBV, Dobrava virus; SANGV, Sangassou virus; SEOV, Seoul virus; MUJV, Muju virus; PUUV, Puumala virus; TULV, Tula virus; LUXV, Lúxī virus; SNV, Sin Nombre virus; ANDV, Andes virus.

The present study proposes that *A. chejuensis*-borne HTNV is a genetically distinct genotype due to the clear natural host species classification and co-divergent evolution of HTNV and *Apodemus spp.* after the geographic segregation between Jeju Island and the Korean peninsula. The species demarcation criterion of the International Committee for Taxonomy of Viruses (ICTV) is currently a 7% difference in amino acid sequences of $G_N/G_C$ and N proteins. The differences in amino acid sequences of the RdRp, glycoproteins, and N proteins of HTNV strains harbored by *A. chejuensis* and *A. agrarius* did not exceed 5%. Taken together, *A. chejuensis*-harbored HTNV is considered *Hantaan orthohantavirus* species, including *A. agrarius*-borne HTNV and *A. peninsulae*-borne SOOV (closely related to AMRV). Continued surveillance and large-scale investigations should be conducted to understand the evolutionary diversification and geographic distribution of *orthohantavirus* and their natural reservoir hosts in ROK.

Rodent-borne orthohantaviruses pose a critical public health threat worldwide. In the ROK, approximately 400 HFRS cases are reported annually, with 33 mortality cases over a decade [7]. Approximately 70% HFRS cases are caused by *A. agrarius*-borne HTNV and the remaining by SEOV carried by *R. norvegicus* or other unidentified agents. Jeju Island includes urban and rural regions with a population of approximately 700,000. It is a popular tourism destination for international tourists to be listed as a Biosphere Reserve (2002), World Natural Heritage Site (2007), and Global Geopark (2010). Overall, visitation to the site has been sustainably increased by an influx of international tourists due to global reputations [38]. The island has been an endemic HFRS region since the first clinical case was officially identified in 2005. After five HFRS cases were recorded from 2001 to 2009, 18 clinical cases have been reported in the recent decade on Jeju Island [7]. Despite the distribution of the rodent reservoir population, HTNV and SEOV have not yet been detected on Jeju Island. In the present study, the genetically distinct HTNV was detected in *A. chejuensis* captured on Jeju Island. Phylogenetic analyses of orthohantaviruses delineated a clear molecular epidemiological link between the patients with HFRS and their reservoir hosts collected at the infection sites [39]. In particular, active surveillance with targeted rodent trapping in the sites suspected of orthohantavirus outbreaks associated with HFRS allowed for phylogenetic distinction within a relatively short distance (about 5 km) to track the precise infectious agents and potential exposed sites [14]. Our previous studies demonstrated a phylogenetic relationship of partial HTNV M segment genome between four US soldiers diagnosed with HFRS and HTNV-positive *A. agrarius* collected at the very likely infection sites [40]. The phylogenetic analysis showed that HTNV derived from *A. chejuensis* genetically linked CUH15-126, which is the only hantaviral sequence collected from the patient with HFRS in southern ROK (Jeollanam Province) [41]. These results indicate that *A. chejugensis*-borne HTNV may be a potential etiological agent in clinical HFRS cases in southern ROK.

The subdivision of DOBV genotypes (Dobrava, Kurkino, Saaremaa, and Sochi) was differentiated by the characteristics in their phylogeny, rodent reservoirs, and geographical distribution [42]. Despite their high genetic homology, different DOBV genotypes induce HFRS with varied severities. The case-fatality rate (CFR) of clinical cases due to the Dobrava genotype harbored by *A. flavicollis* was 10–12% [43–45]. The CFR of HFRS cases caused by the Sochi genotype harbored by *A. ponticus* was approximately 6% [46]. Furthermore, the CFR range for orthohantavirus outbreaks associated with the Kurkino genotype carried by *A. agrarius* was 0.3–0.9% in European Russia. *A. agrarius*-borne SAAV genotype infection appears to be subclinical [47,48]. Thus, the transmission, pathogenicity, and disease risk assessment of HFRS by distinct *A. chejugensis*-borne HTNV in humans should be determined.

Genomic reassortment plays an important role in the evolutionary mechanism by which segmented RNA viruses may shuffle genome segments to generate new viral strains [49]. Genetic exchange of hantaviruses occurs more frequently between genetically closed strains

than between distant strains. The incongruence of phylogenetic patterns in the tripartite segments showed the possibility of genetic reassortment among hantaviruses [46,50]. In this study, the phylogenies of *A. chejuensis*-borne HTNV L, M, and S segments revealed differing levels of incongruence in phylogenetic positions, which indicates the differential evolution of each segment. The L segment of the HTNV shared common ancestors with other HTNV lineages collected from mainland ROK and China. The M segment formed a clade with HTNV strains in the ROK. The S segment showed the independent geographic and distinct cluster with *Hantaan orthohantavirus*, including HTNV, SOOV, and AMRV. The phylogenetic pattern of *A. chejuensis*-borne HTNV may be a configuration compatible with genetic reassortment. However, recombination and reassortment were considered insignificant by RDP4 and GiRaF, respectively. To better understand the complexity of orthohantavirus evolution, future studies should focus on continuous sample collection from reservoir hosts and analysis of hantaviral sequences.

Sexual maturity and aggression are associated with viral transmission among natural host reservoirs [51,52]. The prevalence of orthohantaviruses in rodent populations plays a critical role in understanding hantaviral disease in HFRS-endemic areas. The seroprevalence of orthohantaviruses in rodents demonstrated a higher prevalence of infection in males than females [16,53–57]. Hinson et al. reported that wounded male rats were more frequently infected with SEOV than non-wounded male rats [58]. In this study, the prevalence of HTNV antibodies and RNA was higher in males (7/35, 20%; 5/7, 71.4%, respectively) than those in females (2/29, 6.9%; 1/2, 50%, respectively). Our results showed a gender-specific prevalence of HTNV-infected *A. chejuensis*, supporting that males may contribute to the spread of viral infection via competition with other rodents. However, the number of samples (n = 64) was limited in this study. Thus, continuous large-scale epidemiological surveillance should be conducted to clarify the pattern of male-preferential prevalence of HTNV-infected *A. chejuensis*.

In conclusion, a genetically distinct HTNV was found in *A. chejuensis* captured on Jeju Island, ROK. The phylogeny of orthohantaviruses and their rodent hosts demonstrated that *A. chejuensis* may be infected with ancestral HTNV prior to geological isolation. Our results suggest that *A. chejuensis*-borne HTNV may be a potential etiological agent of HFRS in southern ROK. The discovery of a novel HTNV genotype provides important insights into the evolutionary history, genetic diversity, gender-specific prevalence of orthohantaviruses and a clue to account for HFRS cases that cannot be attributed to *A. agrarius*-borne HTNV infection in the ROK.

## Methods

### Ethics statement

Small mammals were handled in accordance with the Ethical Guidelines of the Korea University Institutional Animal Care and Use Committee (KUIACUC #2019-4 and 2019-171). Captured rodents and shrews were euthanized via cardiac puncture under alfaxalone-xylazine anesthesia. The experiment was performed in an animal biosafety level 3 facility at Korea University.

### Sample collection

Small mammals were captured using Sherman live traps (8 × 9 × 23 cm; H. B. Sherman; Tallahassee, FL, USA) on Jeju Island, ROK from 2018 to 2020. A total of 100 traps were placed in unmanaged grasses, herbaceous vegetation, farmlands, mountains, fields, and forests for 1–3 days (**S5 Table**). The captured small mammals were identified morphologically and transported to Korea University in accordance with the safety guidelines. Multiple tissues of the

rodents and shrews were dissected aseptically and the sera were separated by centrifugation for 5 min at 4˚C. The tissue samples were stored at −80˚C until use.

## Mitochondrial DNA (mtDNA) analysis

Total DNA was extracted from liver tissues using a High Pure PCR Template Preparation Kit (Roche; Basel, Switzerland). The cytochrome *b* gene of mtDNA was amplified using universal primers [59]. The sequences were deposited in GenBank (accession numbers: MW219774-MW2197749).

## Indirect immunofluorescence antibody (IFA) test

The sera and heart fluids of small mammals were used to detect anti-HTNV immunoglobulin G (IgG). The sera or heart fluids were diluted 1:32 and 1:2, respectively, in phosphate buffered saline (PBS) and added to acetone-fixed Vero E6 cells infected with HTNV. The slide was incubated at 37˚C for 30 min. The cells were washed with PBS and distilled water (D.W.). The slides were treated with fluorescein isothiocyanate-conjugated anti-mouse (for *A. chejuensis* and *T. triton*) or mouse/rat (for *C. shantungensis*) IgG (ICN Pharmaceuticals; Laval, Quebec, Canada). The plates were incubated at 37˚C for 30 min and washed with PBS and D.W. The virus-specific fluorescence was evaluated using a fluorescent microscope (Axio Scope; Zeiss; Berlin, Germany).

## Reverse transcription-polymerase chain reaction (RT-PCR)

Total RNA was extracted from lung tissues of rodents and shrews with TRI Reagent Solution (Ambion; Austin, Texas, USA). cDNA was synthesized from 1 µg of total RNA using the High Capacity RNA-to-cDNA kit (Applied Biosystems; Foster City, CA, USA) with OSM55 (5′-TAG TAG TAG ACT CC-3′). The PCR conditions were previously described [18]. All oligonucleotide primer sequences used in this study are shown in **S6 Table**.

## Quantitative PCR (qPCR)

qPCR was performed in a 10 µL reaction mixture containing 1 µg of total RNA. The cycling conditions consisted of denaturation at 95˚C for 10 min, followed by 40 cycles at 95˚C for 15 s, and 60˚C for 1 min using SYBR Green PCR Master Mix (Applied Biosystems) on a QuantStudio 5 Real-Time PCR System (Applied Biosystems).

## Multiplex PCR-based next-generation sequencing (NGS)

cDNA was amplified using HTNV-specific primer mixtures and Solg 2X Uh-Taq PCR Smart mix (Solgent; Seoul, Republic of Korea) according to the manufacturer's instructions. The composition consisted of 12.5 µL 2× Uh pre-mix, 2.0 µL each primer mixture, 10 µL cDNA template, and 10.5 µL D.W. in 25 µL reaction mixture. The PCR cycling conditions were previously described [18]. The PCR amplicons were prepared using the TruSeq Nano DNA LT sample preparation kit (Illumina; San Diego, USA) according to the manufacturer's instructions and previously described study [60]. NGS was performed on a MiSeq benchtop sequencer (Illumina) with 2 × 150 bp using a MiSeq reagent kit v2 (Illumina). The output files were imported and analyzed by CLC Genomics Workbench version 7.5.2 (CLC Bio; Cambridge, MA, USA). The sequences of HTNV strains were deposited in GenBank (accession numbers: MW219756- MW219773).

### Rapid amplification of cDNA ends (RACE) PCR

To obtain the termini 3′ and 5′ genome sequences of HTNV, RACE PCR was conducted using a 3′- and 5′- RACE System for Rapid Amplification of cDNA Ends, Version 2.0 (Invitrogen; Carlsbad, CA, USA), according to the manufacturer's specifications.

### Phylogenetic analysis

The tripartite genomic sequences of HTNV were aligned using the Clustal W algorithm in Lasergene version 5 (DNASTAR Inc.; Madison, WI, USA). For nearly full-length HTNV, phylogenetic analysis was performed using the best-fit GTR+G+I (for L and M segments) and GTR+G (for S segment) substitution models of evolution by the maximum-likelihood (ML) method in MEGA 7. Partial phylogenetic trees were generated by the ML method under the optimal evolutionary models T92+I (for L segment) and T92+G (for M and S segments). The topologies were assessed by bootstrap analysis for 1,000 iterations.

### Co-divergence analysis

A tanglegram algorithm was generated to compare different phylogenetic links matching between HTNV strains and their hosts. The auxiliary lines in the center connect between the phylogenetic trees. The complexity between dendrograms of phylogenies was reduced to the maximum before the full reconciliation analysis. The method was implemented in the R package (dendextend) [61].

### Genetic reassortment analysis

Alignment analysis of the concatenated HTNV L, M, and S segment open reading frame regions were performed using RDP, GENECONV, MAXCHI, CHIMAERA, 3SEQ, BOOTSCAN, and SISCAN in the RDP4 software [62]. A p-value of recombination and reassortment events under 0.05 was considered statistically significant. Genetic events were likely to indicate the occurrence of a genetic exchange when p-value was under 0.05, and the RDP recombination consensus score (RDPRCS) was between 0.4 and 0.6. All parameters of RDP4 were left at the default.

The GiRaF was performed to confirm reassortment events [63]. Nucleotide alignments of the tripartite segments were used as an input source for MrBayes [64]. The best nucleotide substitution models were determined by using MEGA 7. As input for the software, 1,000 unrooted candidate trees were estimated using the GTR+G+I substitution model, the burn-in 50,000 iterations (25%), and sampling every 200 iterations. These trees were used to simulate the phylogenetic uncertainty for segments; the parameters of the GiRaF were default settings. The process was repeated ten times, with ten independent MrBayes based tree data per segment.

### N-glycosylation site analysis

For N-glycosylation site prediction, the full-length amino acid sequences were submitted to the NetNGlyc 1.0 server (https://services.healthtech.dtu.dk) and the parameter was set to default [65].

## Supporting information

**S1 Fig. A phylogenetic tree based on the mitochondrial DNA cytochrome *b* gene of *Apodemus spp*.** Jeju striped field mice (*A. chejuensis*) from Jeju Island were confirmed by conventional polymerase chain reaction (PCR) for the mitochondrial DNA cytochrome *b* gene (coordinates 119–1,080 nt). The phylogenetic tree was generated by the maximum likelihood method using MEGA 7.0. A bootstrap support value >70% is described at the nodes.

*Apodemus* mice are color-coded according to geographic regions (red, mainland ROK; blue, Jeju Island of ROK; green, China).
(TIF)

**S2 Fig. Detection of immunoglobulin G (IgG) antibodies against Hantaan virus (HTNV) from *Apodemus chejuensis* sera and heart fluids by indirect immunofluorescence assay (IFA).** HTNV-infected Vero E6 cells were fixed on each slide. The sera (1:32 dilution) or heart fluids (1:2 dilution) of rodents were used to detect anti-HTNV IgG by IFA, Ac18-6, Ac18-17, Ac19-6, Ac20-30, and Ac20-31; heart fluids of Ac20-5 and Ac20-6. Positive and negative controls used were serum (Aa18-185) and phosphate buffered saline (PBS), respectively. Ac, *A. chejuensis*; Aa, *A. agrarius*.
(TIF)

**S3 Fig. Prediction of N-glycosylation sites in full-length glycoproteins of HTNV and other representative hantaviruses.** The N-linked glycosylation sites for the entire length of glycoproteins in hantaviruses were predicted using NetNGlyc 1.0 server (DTU Bioinformatics), and the parameter was set to default. The potential N-linked glycosylation sites for rodent-borne orthohantaviruses (HTNV, SOOV, and SEOV) are shown. HTNV, Hantaan virus; SOOV, Soochong virus; SEOV, Seoul virus.
(TIF)

**S1 Table. Summary of mapped reads and depth of coverages by Hantaan virus (HTNV) multiplex PCR-based next-generation sequencing.**
(PDF)

**S2 Table. Percentage similarity based on L segment of HTNV nucleotide and amino acid sequences between HTNV from Jeju Island and representative rodent-borne orthohantaviruses.**
(PDF)

**S3 Table. Percentage similarity based on M segment of HTNV nucleotide and amino acid sequences between HTNV from Jeju Island and representative rodent-borne orthohantaviruses.**
(PDF)

**S4 Table. Percentage similarity based on S segment of HTNV nucleotide and amino acid sequences between HTNV from Jeju Island and representative rodent-borne orthohantaviruses.**
(PDF)

**S5 Table. Topography and GPS coordinates of the trapping sites.**
(PDF)

**S6 Table. Oligonucleotide primer sequences generated in this study.**
(PDF)

## Acknowledgments

We thank Dr. Man-Seong Park for supporting experiments.

## Author Contributions

**Conceptualization:** Kyungmin Park, Won-Keun Kim, Jin-Won Song.

**Data curation:** Kyungmin Park, Won-Keun Kim.

**Formal analysis:** Kyungmin Park, Won-Keun Kim, Seung-Ho Lee.

**Funding acquisition:** Jin-Won Song.

**Investigation:** Kyungmin Park, Won-Keun Kim.

**Methodology:** Kyungmin Park, Won-Keun Kim, Seung-Ho Lee, Jongwoo Kim, Jingyeong Lee.

**Resources:** Kyungmin Park, Seung-Ho Lee.

**Supervision:** Jin-Won Song.

**Validation:** Seung-Ho Lee, Jongwoo Kim, Jingyeong Lee, Seungchan Cho, Geum-Young Lee, Jin Sun No, Keun Hwa Lee.

**Visualization:** Kyungmin Park, Won-Keun Kim.

**Writing – original draft:** Kyungmin Park, Won-Keun Kim.

**Writing – review & editing:** Jin-Won Song.

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
