## [Decision Letter · Decision Letter 0]

19 Mar 2021

Dear Dr. Song,

Thank you very much for submitting your manuscript "A novel genotype of Apodemus chejuensis-borne Hantaan Orthohantavirus as a potential etiologic agent of Hemorrhagic Fever with Renal Syndrome in Republic of Korea" for consideration at PLOS Neglected Tropical Diseases. As with all papers reviewed by the journal, your manuscript was reviewed by members of the editorial board and by several independent reviewers. The reviewers appreciated the attention to an important topic. Based on the reviews, we are likely to accept this manuscript for publication, providing that you modify the manuscript according to the review recommendations. 

Sincerely,

Brett M. Forshey

Associate Editor

Emma Wise

Deputy Editor

Kyungmin Park and co-authors analyze the prevalence and diversity of hantaviruses in rodents on Jeju Island, identifying a HTNV in Apodemus chejuensis that is genetically distinct from other HTNV in other rodents. However, this HTNV was genetically related to a sequence from a patient from southern ROK, which suggests that A. chejuensis-associated HTNV might be the cause of HFRS. This is a well-written manuscript, and the analysis and interpretation is solid. The reviewers had mostly minor comments. In addition to the comments from the reviewers below, please consider the following additional minor suggestions:

Abstract, line 37: "The HTNV genomic sequences were phylogenetically related to a viral sequence from the patients with HFRS in southern ROK" - wouldn't this be more precise to say "patient" instead of "patients" since there was only one sequence available?

Abstract, lines 89-90: "We also highlight preemptive precautions for preventing and controlling hantaviral outbreaks to physicians on Jeju Island" I don't recall seeing much in the way of preemptive precautions in the manuscript - this may need to be drawn out a bit more in a couple of sentences in the Discussion.

Results, lines 121-122: "Both IFA+PCR− and IFA−PCR− groups showed that the Ct value was 40, indicating that HTNV RNA was not detectable in tissues." Was the Ct value 40, or not detectable? If HTNV RNA was not detectable, then the Ct value wouldn't be 40, more likely there would be no Ct value to mention. Please clarify here and on Figure 2.

Results, lines 149-150: "The nucleotide similarity of the HTNV L, M, and S segments from Jeju Island compared to those of HTNV 76118 showed 82.2–83.1%, 87.2–88.8%, and 84.8–85.4% homology, respectively" Please check on the use of the terms "similarity" and "homology" in this sentence - would "identity" be more appropriate for nucleotide comparisons?

Discussion, lines 263-264: "Taken together, the distinct genotype of A. chejuensis-harbored HTNV is considered a single species of A. agrarius-borne HTNV" The use of genotype and species in this sentence seem at odds with one another - it's not clear that "species" is the right use of the word in this context. On a similar note, please see the comment from a reviewer below about the use of the term "strain" - consider whether "sequence" or some other term might be more technically precise.

Discussion, lines 274-275: So, from the 18 clinical HFRS cases on Jeju Island over the past decade, there were no sequences available for comparison?

Discussion, lines 305-308: "The phylogenetic pattern of A. chejuensis-borne HTNV may be a configuration compatible with genetic reassortment. However, recombination and reassortment were considered insignificant by Recombination Detection Program 4 and Graph-incompatibility-based Reassortment Finder, respectively (data not shown)." I think it would be useful to include these tools in the Methods, particularly for the "reassortment finder" program, since it gave a result that is not obvious from the phylogenetic tree.

Reviewer's Responses to Questions

**Key Review Criteria Required for Acceptance?**

**Methods**

-Are the objectives of the study clearly articulated with a clear testable hypothesis stated?

-Is the study design appropriate to address the stated objectives?

-Is the population clearly described and appropriate for the hypothesis being tested?

-Is the sample size sufficient to ensure adequate power to address the hypothesis being tested?

-Were correct statistical analysis used to support conclusions?

-Are there concerns about ethical or regulatory requirements being met?

Reviewer #1: YES, the study and it's design was appropriate to support the findings.

Reviewer #2: -Are the objectives of the study clearly articulated with a clear testable hypothesis stated?

Almost yes. 

-Is the study design appropriate to address the stated objectives?

Yes,

-Is the population clearly described and appropriate for the hypothesis being tested?

L 260: Species Demarcation criterion stated here seems to be outdated. Please check for more recent criterion in ICTV using concatenated amino acid sequences of N and Gn/Gc proteins of hantaviruses to demarcate the species. I agree novel sequences from Cheju islands were a novel "genotype" of Hantaan virus. But you should show results with new criteria.

-Is the sample size sufficient to ensure adequate power to address the hypothesis being tested?

Yes

-Were correct statistical analysis used to support conclusions?

Why did you ignore Niviventer borne-Hantaan viruses in this manuscript? I can see the relationship of novel genotype and Amur viruses. In phylogenetic analyses, please include Niviveter borne-viruses. You discuss about reassortment events. For this purpose, More Chinese Hantaan viruses should be included in phylogeny.

-Are there concerns about ethical or regulatory requirements being met?

Yes

Reviewer #3: see final comments.

**Results**

-Does the analysis presented match the analysis plan?

-Are the results clearly and completely presented?

-Are the figures (Tables, Images) of sufficient quality for clarity?

Reviewer #1: The results and representations were correlatable with the objectives of the experiment but there are certain area which the author needs to clarify in order for better representation.

Reviewer #2: -Does the analysis presented match the analysis plan?

Yes

-Are the results clearly and completely presented?

-Are the figures (Tables, Images) of sufficient quality for clarity?

Reviewer #3: see final comments.

**Conclusions**

-Are the conclusions supported by the data presented?

-Are the limitations of analysis clearly described?

-Do the authors discuss how these data can be helpful to advance our understanding of the topic under study?

-Is public health relevance addressed?

Reviewer #1: yes.

Reviewer #2: -Are the conclusions supported by the data presented?

Yes

-Are the limitations of analysis clearly described?

Yes

-Do the authors discuss how these data can be helpful to advance our understanding of the topic under study?

Yes

-Is public health relevance addressed?

It has been reported that mice derived from Cheju island were used to isolate prototype Hantaan virus strain 76118 virus by Dr. H. W. Lee (Hantavirus Hunting). At that time, they explained that this island was HFRS patient-free, so that they used apodemus mice derived from Jeju island. But you clearly showed existence of Hantaan virus in apodemus mice in this island. You have to discuss about following for public health problem?

Is there any Apodemus agrarius genotype in this island? 

Is it possible that the Jeju-derived genotype is less pathogenic?

Since when has this virus been in this island? Host mouse was subspecies of A. agrariu. Rescently, it was considered independent species. Anyway, it was endemic only Cheju island. You have to clearly explain in explanation in Fig. 5

Reviewer #3: see final comments.

**Editorial and Data Presentation Modifications?**

Reviewer #1: Minor revision

Reviewer #2: L 34: The sentence should better to be written as “The prevalence of anti-HTNV antibodies among rodents was 14.1%.”

L 36: The sentence should better to be written as “The detected HTNV genomic sequences were phylogenetically related to a viral sequence derived from HFRS patients in southern ROK.”

L 60: include the word “genome” before the word segments.

L 67: include “of infected rodents” after the word feces.

Figure 1: Is there any special reason to capture rodents only from Jeju-si part of the island?

Figure 2: Have you tried calculating the copy numbers of RNA in tissues?

Figure 4: The sequences for the phylogenetic analysis have been selected only from northern-most and southern-most regions of ROK. Is there a special reason not to select any sequence originated from central or other parts of the country?

Table 3: 

1. Add “from Jeju island” at the end of the table title.

2. The term “strain” refers to the isolated and established virus. These are basically the sequences detected from rodent tissues. Therefore, it is preferred to use another term instead of “strain” in all the places (especially in abstract) stating it in the manuscript. 

L 319: Is there an explanation for the low sample number even though the trapping was conducted for 2 years using relatively large number of traps? 

L 348: Why low dilutions were used for IFA assay? The background reactions can be quite higher at such a low dilution level.

L 350/351: Why only a single secondary antibody was used to screen three species of rodent serum samples? Is there a cross reactivity to anti-mouse IgG in shrew and T. triton sera? In our experience the said secondary antibody does not react with shrew sera.

L 361: RT-qPCR primers were targeted at M-segment of HTNV. Is there any reason to choose M-segment over S-segment which has more copies in the virus for the quantification?

Reviewer #3: see final comments.

**Summary and General Comments**

Reviewer #1: The study entitled “A novel genotype of Apodemus chejuensis-borne Hantaan Orthohantavirus as a potential etiologic agent of Hemorrhagic Fever with Renal Syndrome in Republic of Korea” appears to be a fascinating approach to identify the novel genotype as a pivotal etiologic agent of HFRS. 

 However, there are certain minor areas in which the author needs to clarify in order to refine the work for better representation.

1. Line no. 25 – Repetition of the word “pulmonary”. Hantavirus cardiopulmonary syndrome (HCP)

2. Line 34 - The expression of sentence is not very clear. What does the author mean to say, should clearly explain?

3. Line 87 - HFRS-Borne or Rodent-borne? HTNV is the causative agent of HFRS. 

• What does the author mean to say by the phrase clinical HFRS-borne HTNV strain?

4. Line 95 – 64 should be replaced with sixty-four.

5. Line 108 - The author has mentioned about the collection of shrews and rodents both for the experiment. But the data related to shrews are missing.

• Was there any seropositive case for shrews as well?

6. Line 124 – What was the estimated cut-off value for the RT-qPCR. Author did not mention about that.

Reviewer #2: Kyungmin Park and colleagues have carried out an important study to reveal a potential HFRS causative agent. The study findings show interesting epidemiological features in the new-found HTNV sequences from A. chejuensis in Jeju island, ROK. Even though the total sample number, seropositivity and RNA detection rate is relatively low, the findings are satisfactory enough to justify the importance of studying the potential host rodent species that can harbor hantaviruses in HFRS endemic regions. The close phylogenetic relatedness of new-found HTNV sequences to the HFRS-patient derived HTNV sequences, is a critical observation that alarms the high possibility of this virus genotype to be a candidate etiological agent of HFRS in southern ROK. Overall study is well designed and well implemented. Methods used by the authors are suitable enough to yield significant results. The objectives of the study are well full filled by these findings.

Reviewer #3: The following review is for: 

“A novel genotype of Apodemus chejuensis-borne Hantaan Orthohantavirus as a potential etiologic agent of Hemorrhagic Fever with Renal Syndrome in Republic of Korea” by Park et al. (PNTD-D-20-02190) wherein the authors tested small mammals on Jeju Island during 2018–2020 for orthohantaviruses and provide support for HTNV-specific host use and evolution. The authors provide a compelling argument to support the claim that A. chejuensis -borne HTNV may be a potential etiological agent of HFRS in southern ROK (Republic of Korea) and that the study would bring awareness among physicians for HFRS outbreaks in southern ROK. Additionally, they give attention to the importance of host divergence and geographic isolation events across genetic and geologic evolutionary time.

First concerns:

Though not necessarily a “major” concern, this reviewer feels it is important to mention that taxonomically, Apodemus chejuensis is most likely the sub-species of Apodemus agrarius chejuensis (see https://www.ncbi.nlm.nih.gov/Taxonomy/Browser/wwwtax.cgi?id=754351 and https://www.iucnredlist.org/species/1888/115057408). However, if A. chejuensis is genetically distinct from mainland A. agrarius, the authors could also claim as a new species? (see lines 97-98 S1 Fig).

There may be some push-back from the evolutionary mammalogist community if this is not mentioned. This can be solved in the introduction of A. chejuensis (line 86). Example: 

ln 86 “. . . orhtohantavirus, harbored by the subspecies A. agrarius chejuensis (hereafter, A. chejuensis) captured on Jeju Island.”

Check throughout to remain consistent with “orthohantavirus” replacing “hantavirus”. (e.g. lns 66, 69,79,80,86, etc.; it’s hard for this reviewer to get used to also!)

Minor concerns:

Title ln2: might consider adding full subspecies epithet.

ln27: “orthohantaviruses”

ln32: same type of introduction of subspecies (if decided to go that route).; not sure “respectively is needed?

ln35: show (n=x/y) for percentages?

ln55: “significant” might consider different word; could imply the need for statistical evidence.

ln64: taxonomic orders do not need italics; change “Soricomorpha” to “Eulipotyphla”

ln81: change “In” to “On”?

ln94-95: suggested re-write: “The trapped small mammals consisted of 91.4% (64/70) A. chejuensis and 2.8% (2/70) T. triton rodents and 2 C. shantugensis shrews.”

ln102-104: Fig 1 and caption, it might be helpful to place a site number in the text and on the figure.

ln106: consider “Trapping results of small mammals collected at sites on Jeju Island in the Republic of Korea, 2018-2020.”

ln107: bottom of table, “no collection” descriptor is likely not necessary 

lns114-115: sentence is a little confusing. It suggests to the reader that the tests (antibodies vs RNA) were exclusive to host and not used for each.

ln135 & 640-641: S1 Table might consider reducing the large number values to 3-4 digits with the conversion metric in column header of the table (e.g. total reads, reads mapped, and depth of coverage).

ln152-153: not sure the last sentence is needed. 

ln156: after “positions 644-648” does this refer to S3 Table? May want to include table reference.

ln257: consider a new paragraph starting with the last “The . . .”

ln257-266: The authors make a very compelling argument, and this reviewer agrees with their conclusions.

ln286-289: Again, the authors have provided important information for the medical community as stated in their claims.

ln313: sentence is a little confusing; suggest changing to: “The seroprevalence of orthohantaviruses in rodents demonstrated a higher prevalence of infection in males than females.”

PLOS authors have the option to publish the peer review history of their article (what does this mean?). If published, this will include your full peer review and any attached files.

Reviewer #1: No

Reviewer #2: No

Reviewer #3: Yes: Matthew T Milholland

Figure Files:

Data Requirements:

Reproducibility:

References

---

## [Editor Report · Decision Letter 1]

22 Apr 2021

Dear Dr. Song,

We are pleased to inform you that your manuscript 'A novel genotype of Hantaan Orthohantavirus harbored by Apodemus agrarius chejuensis as a potential etiologic agent of Hemorrhagic Fever with Renal Syndrome in Republic of Korea' has been provisionally accepted for publication in PLOS Neglected Tropical Diseases.

Best regards,

Brett M. Forshey

Associate Editor

Emma Wise

Deputy Editor

---

## [Editor Report · Acceptance letter]

7 May 2021

Dear Dr. Song,

We are delighted to inform you that your manuscript, "A novel genotype of Hantaan Orthohantavirus harbored by Apodemus agrarius chejuensis as a potential etiologic agent of Hemorrhagic Fever with Renal Syndrome in Republic of Korea," has been formally accepted for publication in PLOS Neglected Tropical Diseases.

Best regards,

Shaden Kamhawi

co-Editor-in-Chief

Paul Brindley

co-Editor-in-Chief
